# Expression of divergent methyl/alkyl coenzyme M reductases from uncultured archaea

Nana Shao [1], Yu Fan[2], Chau-Wen Chou[3], Shadi Yavari[4], Robert V. Williams[3], I. Jonathan Amster [3], Stuart M. Brown[5], Ian J. Drake[6], Evert C. Duin[4], William B. Whitman [1✉] & Yuchen Liu [5✉]

Methanogens and anaerobic methane-oxidizing archaea (ANME) are important players in the global carbon cycle. Methyl-coenzyme M reductase (MCR) is a key enzyme in methane metabolism, catalyzing the last step in methanogenesis and the first step in anaerobic methane oxidation. Divergent *mcr* and *mcr*-like genes have recently been identified in uncultured archaeal lineages. However, the assembly and biochemistry of MCRs from uncultured archaea remain largely unknown. Here we present an approach to study MCRs from uncultured archaea by heterologous expression in a methanogen, *Methanococcus maripaludis*. Promoter, operon structure, and temperature were important determinants for MCR production. Both recombinant methanococcal and ANME-2 MCR assembled with the host MCR forming hybrid complexes, whereas tested ANME-1 MCR and ethyl-coenzyme M reductase only formed homogenous complexes. Together with structural modeling, this suggests that ANME-2 and methanogen MCRs are structurally similar and their reaction directions are likely regulated by thermodynamics rather than intrinsic structural differences.

[1] Department of Microbiology, University of Georgia, Athens, GA, USA. [2] EMTEC IT, ExxonMobil Technical Computing Company, Annandale, NJ, USA. [3] Department of Chemistry, University of Georgia, Athens, GA, USA. [4] Department of Chemistry and Biochemistry, Auburn University, Auburn, AL, USA. [5] Energy Sciences, ExxonMobil Technology & Engineering Company, Annandale, NJ, USA. [6] Biomedical Sciences, ExxonMobil Technology & Engineering Company, Annandale, NJ, USA. ✉email: whitman@uga.edu; yuchen.liu@exxonmobil.com

Methanogens or methanogenic archaea are considered one of the earliest microbial life forms on Earth[1,2]—and together with anaerobic methane-oxidizing archaea (ANME), they play pivotal roles in the global carbon cycle. Today methanogens produce about one billion tons of methane annually in anoxic environments using various substrates including $H_2$/$CO_2$, formate, acetate, and $C_1$-methylated compounds[3] as well as those recently discovered including coal components[4,5] and long-chain alkanes[6]. In anoxic marine sediments, it is estimated that ~90% of the biogenic methane is oxidized by ANME to $CO_2$ using a reverse methanogenesis pathway[7], mitigating the release of methane into the atmosphere.

All ANME and many methanogens remain uncultured as single-species cultures. Based upon environmental metagenomes and enrichment cultures, ANME are biochemically and genetically closely related to methanogens, sharing a similar set of enzymes for anaerobic methane oxidation (AOM) in the opposite direction of methane formation[8–11]. ANME use methane as a carbon and energy source and transfer electrons from methane to syntrophic sulfate-reducing bacterial partners[9,12,13] or inorganic electron acceptors, such as nitrate[14], Fe(III)[15–17], and Mn(IV)[15]. The known ANME do not constitute a single taxonomic group and belong to the orders "*Ca.* Methanophagales" (ANME-1) and *Methanosarcinales* (ANME-2 and ANME-3)[10]. The *Methanosarcinales* order also contains methanogens. The physiological and biochemical details of ANME remain largely unknown due to the lack of pure cultures and slow growth of enrichments[8,18].

The methyl-coenzyme M (CoM) reductase (MCR) is a key enzyme of anaerobic methane metabolism[19]. It catalyzes the last $CH_4$-formation reaction in methanogenesis and the first $CH_4$-activating reaction in AOM. The reversibility of the MCR reaction (reaction 1) has been demonstrated experimentally with a *Methanothermobacter marburgensis* MCR[20]. Recently the related alkyl-coenzyme M reductase (ACR) was proposed to catalyze the oxidation of short-chain alkanes (e.g., ethane, propane, and butane) by anaerobic alkane-oxidizing archaea (ANKA)[21–24].

$$CH_3 - S - CoM(methyl - coenzyme\ M) + HS$$
$$- CoB(coenzyme\ B) \Longleftrightarrow CH_4 + CoM - S - S - CoB$$
$$(1)$$

The MCR complex is composed of a dimer of heterotrimers $(\alpha\beta\gamma)_2$ with a molecule of the Ni-containing tetrapyrrole coenzyme $F_{430}$ in each of the two active sites[25]. Each $F_{430}$ is deeply buried within the protein complex and only accessible from the outside by a 50 Å channel formed from multiple subunits, McrA, A', B, and G or McrA', A, B', and G'[26,27]. The Ni(I) oxidation state of $F_{430}$ is required for activity. The Ni(II)/Ni(I) couple has an extremely negative redox potential ($E^{o'}$) below −600 mV[28], and therefore MCR is very oxygen sensitive and requires a complex enzyme system for ATP-dependent reductive activation[29]. Multiple unique posttranslational modifications (PTMs) are present in the McrA subunit and fine-tune the MCR stability and activity[30–32]. Although crystal structures of an ANME-1 MCR[33] and an ethyl-coenzyme M reductase (ECR)[34] have been solved, the assembly and biochemical properties of both ANME and ANKA enzymes remain poorly understood. Heterologous expression of the genes encoding an ANME-1 MCR in *Methanosarcina acetivorans* stimulated methane oxidation by the recombinant organism, providing further evidence for the role of these enzymes[35]. Recently, the *Methanothermococcus okinawensis* MCR was heterologously expressed in the model methanogen *Methanococcus maripaludis*[36]. Here, we further developed the heterologous expression of MCRs in *M. maripaludis* that paves the way for studying enzyme complexes from uncultured archaea.

## Results

**MCRs are widespread and diverse in archaea.** Recent environmental genomics studies have revealed many archaeal lineages of potential methanogens and ANME that have not been cultivated to date[10]. Here, we investigated the distribution of MCR homologs across 1070 assembled archaeal genomes from the Genome Taxonomy Database (GTDB) with completeness >80% and contamination <10%. A total of 307 genomes contained all three of the genes (*mcrA*, *mcrB*, and *mcrG*) necessary to encode the MCR subunits (Supplementary Data 1). In the rank-normalized phylogenetic tree based upon all 1070 genomes[37], these *mcr*-containing archaea included methanogens, ANME-1, ANME-2, ANKA, and other archaea of unknown metabolic types and were interspersed with lineages that do not share these genes (Fig. 1). The widespread distribution of *mcr* genes in archaea supports the hypothesis that methane metabolism is an ancient trait likely present in the archaeal root[38–40].

In addition to the structural genes, many *mcr* operons encoded two accessory proteins, McrC and McrD. While the roles of McrC and D are not well characterized, McrC has been shown to participate in the MCR activation complex[29] and McrD may facilitate the addition of coenzyme $F_{430}$ to the complex[41]. Three major *mcr* operon structures were identified, *mcrBDCGA*, *mcrBDGA*, and *mcrBGA*, which possessed a strong phylogenetic signal (Fig. 1). Notably, methanogen and ANME-2 genomes predominantly contained the *mcrBDCGA* and the *mcrBDGA* operons; whereas ANME-1 genomes possessed the shorter *mcrBGA* operon with *mcrC* at a separate locus (Fig. 1). The ANKA genomes mainly possessed one or more *mcrBGA* and/or *mcrBAG* operons. The lack of *mcrD* homologs in ANME-1 and ANKA genomes may be emblematic of other major differences with the enzymes from methanogens. For instance, they contain modified nickel-containing $F_{430}$ cofactors, e.g., thiomethylated $F_{430}$ from an ANME-1 MCR[33] and dimethylated $F_{430}$ from *Candidatus* Ethanoperedens thermophilum MCR[34].

Gene duplications of *mcr* are common in archaea. Among methanogens, many genomes from the *Methanobacteriales*, *Methanococcales*, *Methanomicrobiales* orders have one copy of *mcrBDCGA* and a second copy of either *mcrBDGA* or *mcrBGA* (Supplementary Data 1). In *M. marburgensis*, the two MCR isoenzymes are differentially expressed depending on $H_2$ concentrations[42–44], suggesting that *mcr* duplications may play a role in physiological acclimations to varied growth conditions. On the other hand, the proposed ANKA genomes often have multiple *mcrBGA/BAG* operons[21,45,46], suggesting that *mcr* duplications may have expanded its function from methane to multi-carbon alkane metabolism.

**Optimization of heterologous expression of MCRs.** The heterologous expression of MCRs in *M. maripaludis* was optimized systematically. First, a constitutive histone promoter (P*hmvA*)[36] was compared with a recently developed phosphate-dependent promoter (P*pst*)[47], which initiates expression upon phosphate limitation and partially separates expression from growth. A Flag-Strep₂ tag was added to the N-terminus of McrG from the *Methanococcus aeolicus* *mcrBDCGA* operon (Fig. 2a). Based upon Western blotting from a previous study, the P*hmvA* and P*pst* promoters yielded *M. aeolicus* MCR (MCR$_{aeo}$) of 2.4% and 5.8% of total protein, respectively[47]. Thus, the P*pst* promoter was superior and utilized in subsequent experiments. Second, the effect of tag locations was examined. The Flag-Strep₂ tag was added to the N-terminus of McrG (NG), the N-terminus of McrB (NB), or the C-terminus of McrA (CA) of MCR$_{aeo}$. The identities of purified proteins were confirmed by mass spectrometry following SDS-PAGE (Fig. 2b). Small amounts of McrD were

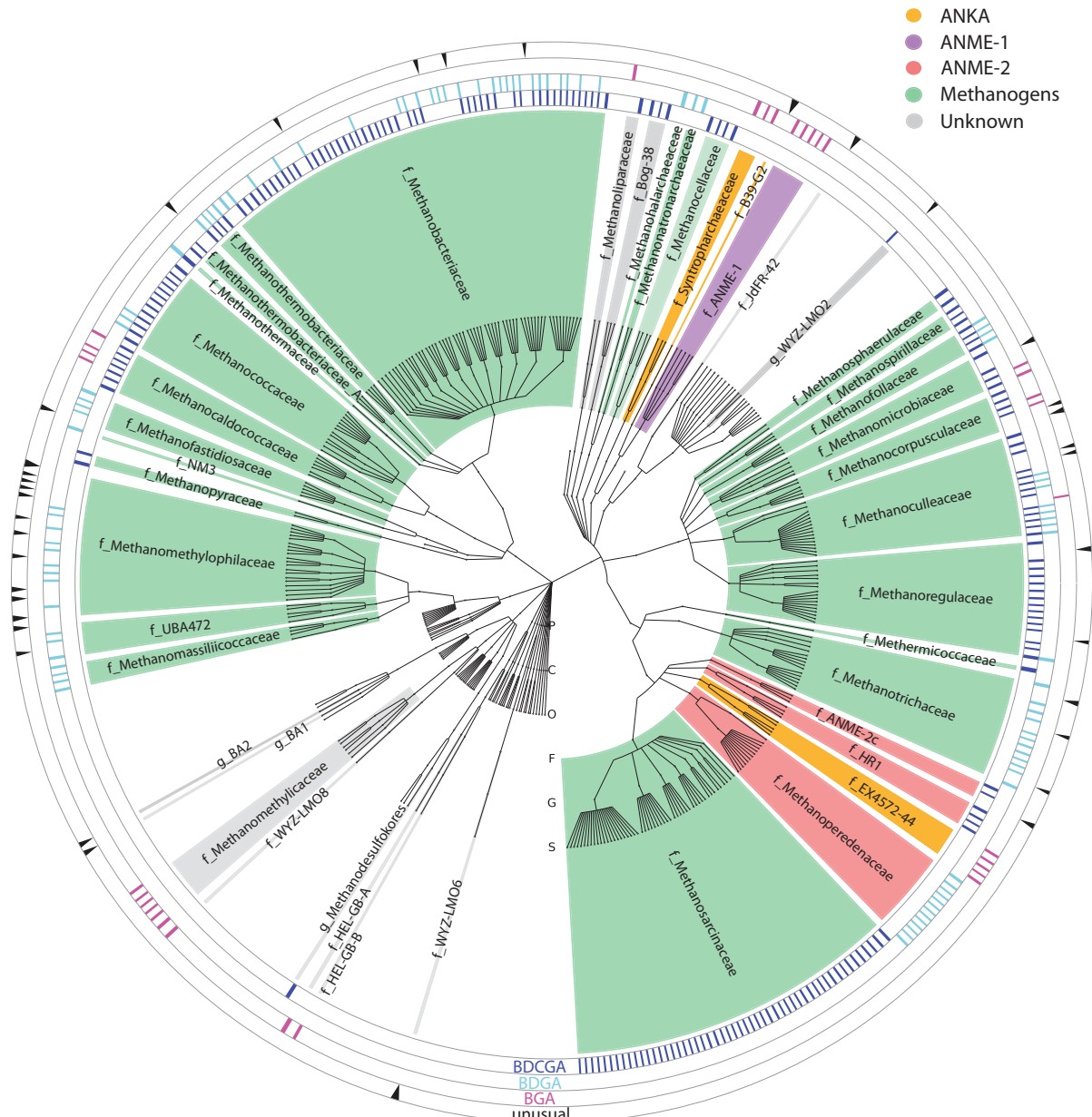

**Fig. 1 Distribution and gene clusters of mcr among archaea.** A total of 307 *mcr* genes were identified from 1070 archaeal genomes including methanogens (n = 252), ANME-1 clade (n = 5), ANME-2 (n = 20), ANKA (n = 9), and other archaea with unknown metabolism (n = 21). Accession numbers are given in Supplementary Data 1. Color shading: green, methanogens; purple, ANME-1 archaea; red, ANME-2 archaea; orange, proposed ANKAs; gray, archaea containing *mcr* with unknown functions. The operon structures (*mcrBDCGA*, *mcrBDGA*, or *mcrBGA*) are represented by multiple rings outside the rank-normalized phylogenetic tree. BDCGA (in blue), *mcrBDCGA* operon; BDGA (in cyan), *mcrBDGA* operon; BGA (in magenta), *mcrBGA* operon; unusual (in black), *mcr* genes lack the three recognized common operon structures. Multiple hits of a single genome indicate the presence of multiple copies of *mcr*. Taxonomic classification: P phylum, C class, O order, F family, G genus, S species. Lineages without *mcr* were truncated at the order level.

identified in addition to the McrB, G, A subunits. In all three cases, the MCR$_{aeo}$ yields were ~6% of total cellular proteins, suggesting that tag locations did not affect the protein production level. The ultraviolet-visible (UV–vis) spectra of the purified MCR$_{aeo}$ exhibited a maximal absorption peak at 425 nm, which was typical for the MCR holoenzyme and slightly lower than the absorption maximum of methanol-extracted F$_{430}$ at 430 nm (Fig. 2c). Based on the molar extinction coefficient $\varepsilon_{430nm} = 22{,}500 \, \text{M}^{-1} \text{cm}^{-1}$ and an HPLC-based analysis (Supplementary Fig. S1), the purified MCR$_{aeo}$ with NB or NG tags was fully

assembled with F$_{430}$, whereas the CA tag reduced the F$_{430}$ content by 30% (Fig. 2c). Therefore, tag locations affected F$_{430}$ assembly, and the NB and NG tags were suitable for productions of the holo-MCR. Lastly, the presence of PTMs—including thioglycine, 1-*N*-methylhistidine, 5-(*S*)-methylarginine, and 2-(*S*)-methylglutamine—of the recombinant McrA$_{aeo}$ were identified by liquid chromatography-tandem mass spectrometry (LC-MS/MS) (Supplementary Table 1). This indicated that our heterologous expression system resulted in the same PTMs as found for the *M. maripaludis* and closely related *M. okinawenesis* MCRs.

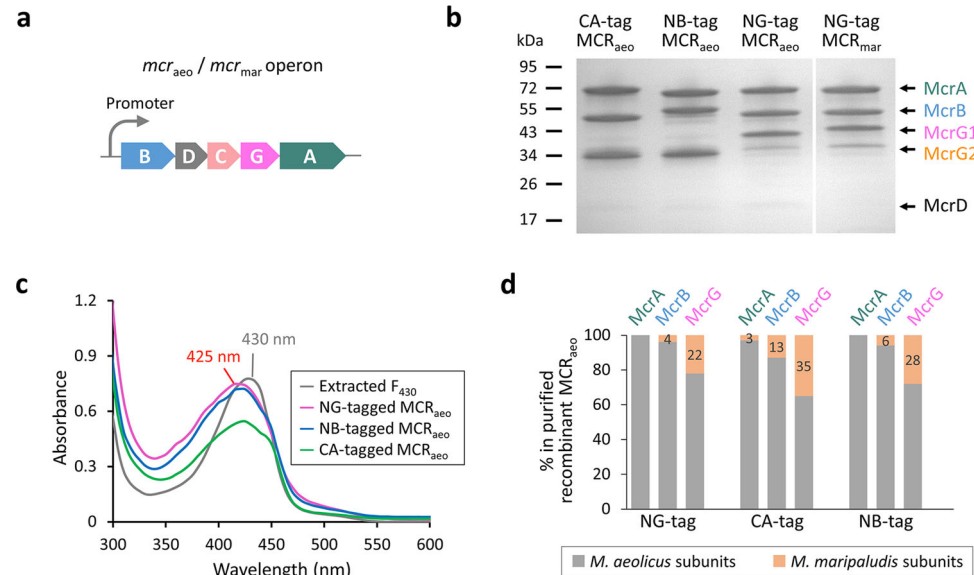

**Fig. 2 Heterologous expression of methanococcal MCRs. a** The *mcr* operon structure of *M. aeolicus* (*mcr*$_{aeo}$) and *M. maripaludis* (*mcr*$_{mar}$). **b** SDS-PAGE analysis of the recombinant MCR$_{aeo}$ and MCR$_{mar}$ purified by Strep-tag affinity and ion-exchange chromatography from *M. maripaludis*. The molecular weights based on standards are labeled on the left. The Flag-Strep$_2$ tag was added to the C-terminus of McrA (CA), the N-terminus of McrB (NB), or the N-terminus of McrG (NG) positions. All subunits were identified by MALDI-TOF MS and labeled on the right. McrG1 and McrG2 represents tagged and untagged McrG, respectively. **c** UV–visible spectra of purified recombinant MCR$_{aeo}$ and MCR$_{mar}$ (all at 7.5 mg mL$^{-1}$ concentration) compared to coenzyme F$_{430}$ extracted from the *M. marburgensis* MCR. **d** The relative abundance of *M. aeolicus* (in gray) vs. host *M. maripaludis* (in orange) MCR in each subunit of the purified recombinant MCR$_{aeo}$ complex determined by LC-MS/MS. The percentages of *M. maripaludis* protein in total protein of each subunit are labeled.

Although the recombinant MCR$_{aeo}$ assembled into a holo-complex, SDS-PAGE showed that the NG-tagged complex contained an extra McrG2 subunit (Fig. 2b). Mass spectrometry identified that McrG1 and G2 were Flag-Strep$_2$-tagged McrG$_{aeo}$ and the untagged host *M. maripaludis* McrG$_{mar}$, respectively. This chimerism was also observed for MCR$_{aeo}$ expressed with other tag positions and the recombinant *M. maripaludis* MCR (Fig. 2b, d). The *M. maripaludis* McrG comprised 30 ± 6% of the total McrG regardless of the tag positions. In contrast to McrG, LC-MS/MS analyses of the recombinant MCR$_{aeo}$ complex found only small amounts of *M. maripaludis* McrA and McrB (Fig. 2d). The chimeric complexes were further characterized by native PAGE and intact protein mass spectrometry (Fig. 3). Two complexes (Complex I and II) were observed for the purified recombinant MCR$_{aeo}$ and MCR$_{mar}$ with native PAGE (Fig. 3a). SDS-PAGE of the two complexes found that they differed in the presence of the extra untagged *M. maripaludis* McrG (Fig. 3b). Intact protein mass spectrometry determined that complexes I and II had molecular masses of 288.4 and 283.8 kDa, respectively (Fig. 3c and Supplementary Table 2). Accordingly, complex I matched a α$_2$β$_2$h$_2$f$_2$ complex, where α, β, h, and f corresponds to *M. aeolicus* McrA, McrB, tagged McrG, and F$_{430}$, respectively. Complex II matched a α$_2$β$_2$hγf$_2$ with γ corresponding to the untagged *M. maripaludis* McrG (Fig. 3d). These results indicated that McrG readily binds McrA and B subunits from a different origin.

**Operon structure affects heterologous expression levels of MCR in *M. maripaludis*.** The accessory proteins McrC and McrD are highly conserved in *mcr* operons of methanogens but often absent in those of ANME and ANKA. To investigate their roles in MCR assembly, truncated *M. aeolicus mcr* operons—including *mcrBDGA*, *mcrBCGA,* and *mcrBGA*—were constructed. In all cases, the Flag-Strep$_2$-tag was added to the NG position.

Expressions of MCR$_{aeo}$ from all three truncated operons yielded complexes similar to those of the full *mcrBDCGA* operon (Fig. 4a, b). UV–vis spectra (Fig. 4c) and HPLC analysis confirmed the full complement of F$_{430}$ in the purified MCR$_{aeo}$. In addition, the major PTMs were also present (Supplementary Table 1). However, the expression levels of truncated *mcr* operons were about threefold lower than that of the full operon (Fig. 4d), although the cause of the reduced protein levels is currently unclear. These results demonstrated that the presence of *mcrCD* inside the *mcr* operon was not necessary for MCR assembly and PTMs.

McrC was proposed to act in *trans*, i.e., co-transcription of *mcrC* with the *mcr* operon was not required for its function, based upon the observation that the *M. marburgensis* McrC co-purified with the MCR activation complex, which was encoded outside the *mcr* operon[29]. To test this hypothesis, two pull-down experiments were performed. First, the *M. aeolicus mcrBDCGA* genes were expressed with a Flag-Strep$_2$ tag at the N-terminal of McrC (McrC$_{aeo}$). The McrA, B, and G subunits of both *M. aeolicus* (transcribed from a plasmid) and the host *M. maripaludis* (transcribed from the genome) co-purified with the tagged McrC$_{aeo}$ (Fig. 5a, c), suggesting that McrC interacts directly with both MCRs independently of co-transcription. Second, the Flag-Strep$_2$-tagged *M. maripaludis* McrC (McrC$_{mar}$) alone was expressed from a plasmid. The tagged McrC$_{mar}$ also purified together with the three MCR subunits expressed from the genome (Fig. 5a). Furthermore, other proteins not from *mcr* operons co-purified with both the tagged McrC$_{aeo}$ and McrC$_{mar}$. These proteins included two previously identified MCR activation complex components (component A2 and methanogenesis marker protein 7)[29] and two other methanogenesis marker proteins 3 and 17 (Fig. 5a).

Two experiments confirmed that McrD functions in *trans*. First, the host *M. maripaludis* McrD was present in the purified recombinant *M. aeolicus* MCRs expressed from both the full and the truncated *mcr* operons lacking *mcrD* (Fig. 4a). Second, the *M.*

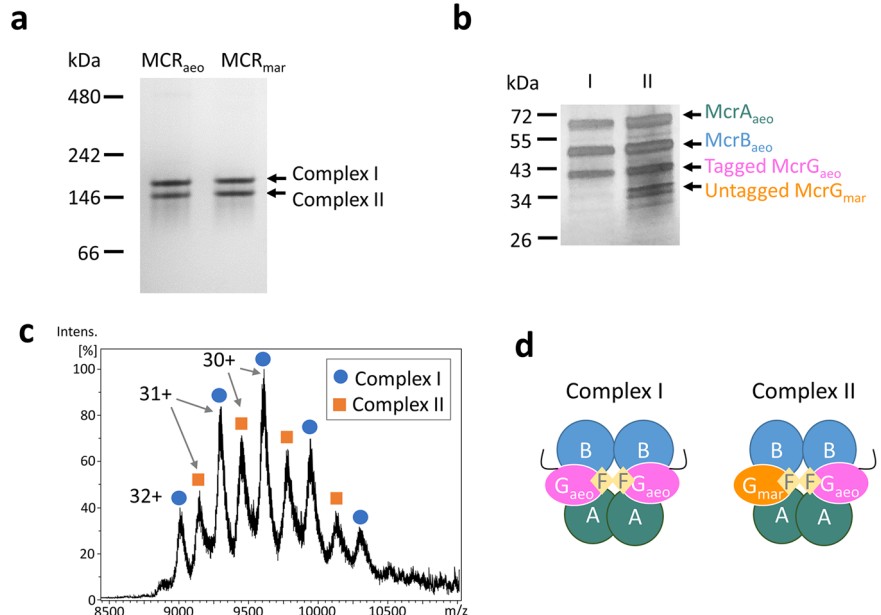

**Fig. 3 Characterization of recombinant complexes. a** Native-PAGE analysis of the purified recombinant $MCR_{aeo}$ and $MCR_{mar}$. Both constructs had a Flag-$Strep_2$ tag added to the N-terminus of McrG. **b** The two complexes of $MCR_{aeo}$ were eluted from native-PAGE gel slices, analyzed by SDS-PAGE, and silver stained. **c** The native molecular masses of the $MCR_{aeo}$ complexes I and II were determined as 288.4 and 283.8 kDa, respectively, by intact protein mass spectrometry. The charges of the peaks are labeled. **d** Models of complexes I and II. A, B, and $G_{aeo}$ represent *M. aeolicus* McrA, McrB, and McrG subunits, respectively. $G_{mar}$ denotes the untagged host *M. maripaludis* McrG. The black line symbolizes the tag. F stands for coenzyme $F_{430}$.

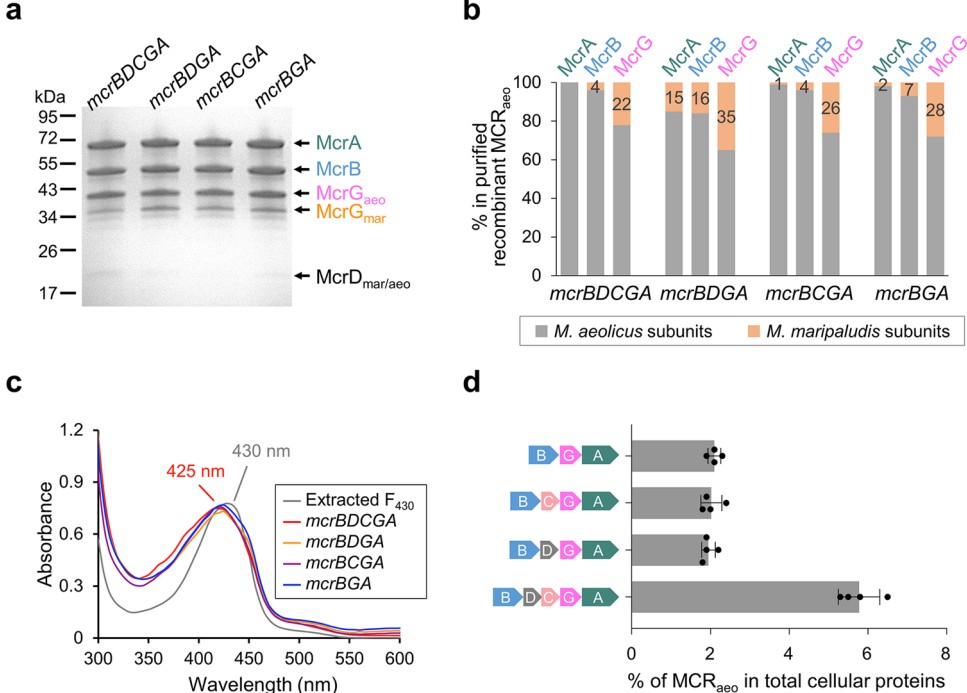

**Fig. 4 Expression of recombinant $MCR_{aeo}$ with truncated operons. a** SDS-PAGE analysis of the recombinant $MCR_{aeo}$ purified by Strep-tag affinity and ion-exchange chromatography. All constructs had a Flag-$Strep_2$ tag added to the N-terminus of McrG. The operon structures are labeled above each lane. The molecular weights based on standards are labeled on the left. All subunits were identified by MALDI-TOF MS and labeled on the right. **b** The relative abundance of *M. aeolicus* (in gray) vs. host *M. maripaludis* (in orange) MCR in each subunit of the co-purified complexes determined by LC-MS/MS. The percentages of *M. maripaludis* protein in total protein of each subunit are labeled. **c** UV–vis spectra of purified $MCR_{aeo}$ (all at 7.5 mg $mL^{-1}$ concentration) compared to coenzyme $F_{430}$ extracted from MCR. **d** Expression levels of recombinant $MCR_{aeo}$ determined by western blotting. Error bars represent the standard deviation of four independent cultures.

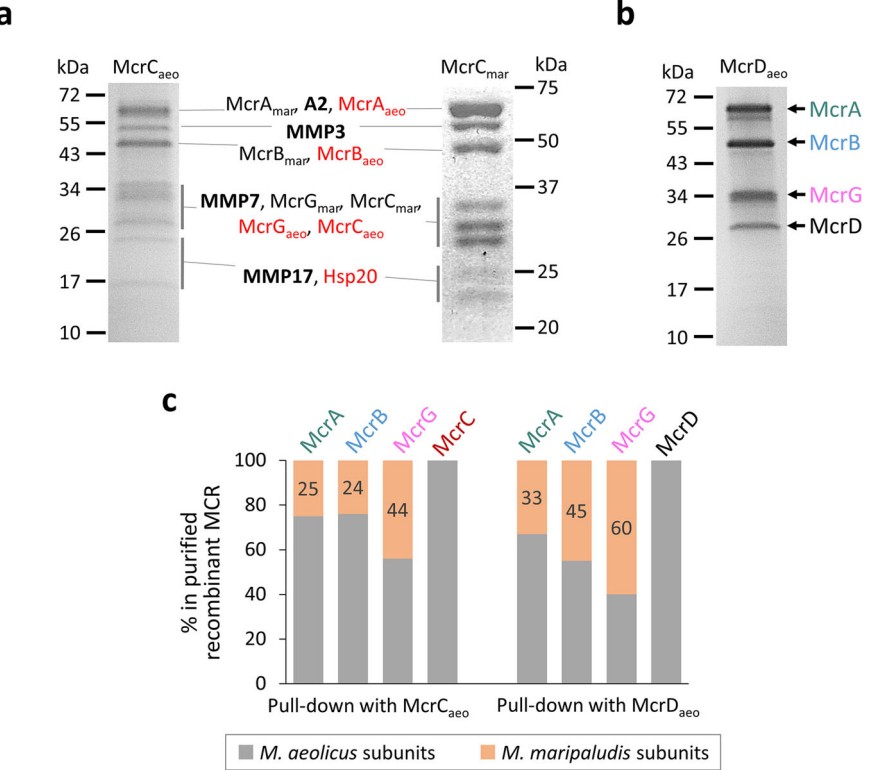

**Fig. 5 Pull-down analyses of accessory proteins McrC and McrD. a** The McrC$_{aeo}$ construct had the full *M. aeolicus mcrBDCGA* operon with the Flag-Strep$_2$ tag added to the N-terminal of McrC. The McrC$_{mar}$ construct contained only the *M. maripaludis mcrC* with an N-terminal Flag-Strep$_2$ tag. Proteins co-purified with McrC$_{aeo}$ and McrC$_{mar}$ were separated by SDS-PAGE and identified by MALDI-TOF MS. Proteins only purified with McrC$_{aeo}$ are labeled in red. Besides MCR subunits, the co-purified proteins (in bold) include *M. maripaludis* A2 protein (locus tag Mmp_0620), MMP3 (methanogenesis marker protein 3, locus tag Mmp_0154), MMP7 (methanogenesis marker protein 7, locus tag Mmp_0421), MMP17 (methanogenesis marker protein 17, locus tag Mmp_0656), and heat shock protein Hsp20 (locus tag Mmp_0684). **b** The McrD$_{aeo}$ construct had the full *M. aeolicus mcrBDCGA* operon with the Flag-Strep$_2$ tag added to the N-terminal of McrD. Proteins co-purified with McrD$_{aeo}$ were separated by SDS-PAGE and identified by MALDI-TOF MS. **c** The relative abundance of *M. aeolicus* (in gray) vs. host *M. maripaludis* (in orange) MCR in each subunit of the purified complexes determined by LC-MS/MS. The percentages of *M. maripaludis* protein in total protein of each subunit are labeled.

*aeolicus* operon encoding a Flag-Strep$_2$ tag at the N-terminal of McrD (McrD$_{aeo}$) was expressed from a plasmid in *M. maripaludis* for a pull-down experiment. All three MCR subunits from both *M. aeolicus* and *M. maripaludis* co-purified with the tagged McrD$_{aeo}$ (Fig. 5b, c), suggesting that McrD$_{aeo}$ interacted with the host MCR expressed from the genome.

**Heterologous expression of ANME MCRs and an ECR**. The robust expression system was applied to produce MCRs from uncultured archaea. Two ANME-1 MCRs, four ANME-2 MCRs, and one ECR were selected for heterologous expression (Supplementary Table 3). In all cases, the Flag-Strep$_2$ tag was added to the N-terminal of McrG under the control of the P*pst* promoter. The temperature was identified as an important factor for ANME MCR and ECR production. At 37 °C, the optimum growth temperature of the host *M. maripaludis*, only low levels of recombinant MCRs were detected by Western blotting (Supplementary Fig. S2) even though the mRNA copy numbers quantified by qRT-PCR suggested a higher level of mRNA for the recombinant ANME-1_G37 MCR than the native host MCR (Supplementary Fig. S3). Given that many ANME metagenomes were obtained from deep sea sediments where the temperatures were near 2 °C, expression at lower temperatures was examined. Following growth close to the temperature minimum of *M. maripaludis* (25 °C), the expression of ANME MCRs and ECR were much improved. For instance, ANME-1_BS and ANME-2b_HR1

represented 1–1.5% of total cellular proteins (Supplementary Fig. S2). These results suggested that the ANME MCRs and ECR were unstable at higher temperatures or susceptible to degradation when expressed in *M. maripaludis*.

The protein compositions of the purified ANME-1_BS MCR (MCR$_{ANME-1\_BS}$), ANME-2b_HR1 MCR (MCR$_{ANME-2\_HR1}$), and *Ca.* Ethanoperedens thermophilum E50 ECR (ECR$_{E50}$) were further studied (Fig. 6a). Mass spectrometry analysis confirmed that the purified MCR$_{ANME-1\_BS}$ and ECR$_{E50}$ comprised all three subunits McrA, B, and G (Fig. 6b, c) without the host *M. maripaludis* MCR. By contrast, the tagged McrG of ANME-2b_HR1 co-purified with the three *M. maripaludis* MCR subunits (Fig. 6d), suggesting that McrG$_{ANME-2\_HR1}$ and MCR$_{mar}$ assembled into a hybrid complex. Protein sequence alignments showed that the McrG of ANME-1 has a C-terminal extension longer than those of methanogens, ANME-2, and *Ca.* E. thermophilum (Supplementary Fig. S4); this extension may inhibit interactions with the host methanococcal MCR.

The structural basis of MCR hybrid complex formation was further analyzed by computational modeling. A homology model of the *M. maripaludis* MCR complex was built with RosettaCM[48] (Supplementary Fig. S5) and used for protein docking with McrG from ANME-1_BS and ANME-2b_HR1 by RosettaDock[49,50] (Fig. 6e–g). The in silico docking of McrG$_{ANME-2\_HR1}$ with *M. maripaludis* MCR was successful with the lowest interface root mean squared deviation (I_rmsd) = 1.848 Å (Fig. 6e). In this model, 10 amino acids of McrG$_{ANME-2\_HR1}$ were identified within

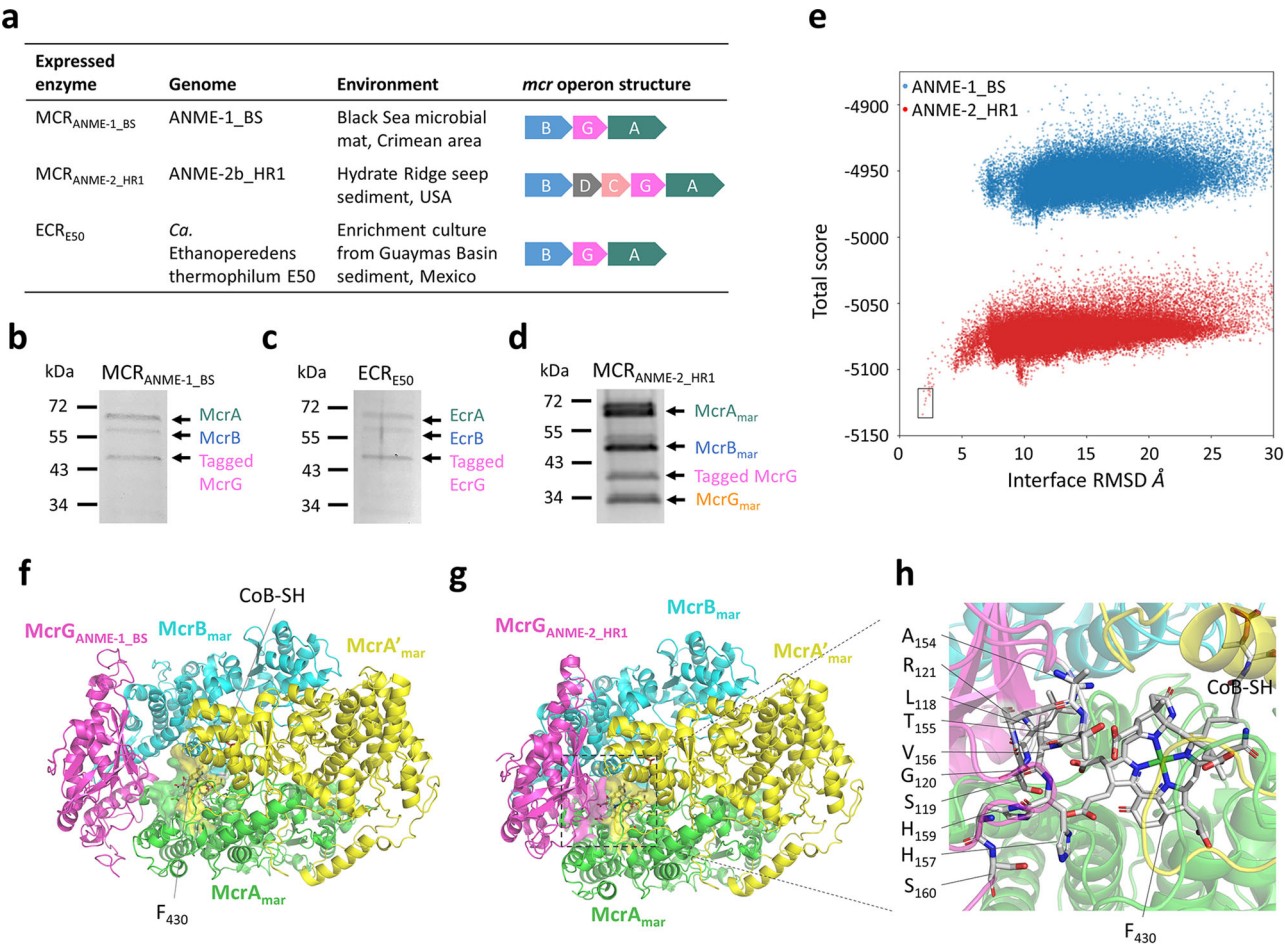

**Fig. 6 ANME MCRs and an ECR expressed in *M. maripaludis*. a** General information and operon structures of MCR_ANME-1_BS, MCR_ANME-2b_HR1, and ECR_E50. **b–d** SDS-PAGE analysis of the purified MCR_ANME-1_BS, ECR_E50, and MCR_ANME-2_HR1 produced from *M. maripaludis* grown at 25 °C. The molecular weights based on standards are labeled on the left. Protein identities (labeled on the right) were confirmed by MALDI-TOF MS. The tagged McrG_ANME-2_HR1 co-purified with the host MCR_mar. **e** Total score (Rosetta energy unit; REU) vs. I_rmsd plot for local docking simulations of McrG_ANME-1_BS (blue) and McrG_ANME-2_HR1 (red) to the *M. maripaludis* McrA, B, G complex. The plot displays 60,000 scoring models. The best model obtained from the McrG_ANME-1_BS docking had a 5.869 Å I_rmsd and a -4965.008 total score. The best model obtained from the McrG_ANME-2_HR1 docking had a 1.848 Å I_rmsd and a -5133.935 total score. The ten lowest-energy scores with I_rmsd < 2.5 Å are labeled in the black box. **f, g** Structural models with the smallest I_rmsd in the simulation of the McrG_ANME-1_BS (magenta, **f**) and McrG_ANME-2_HR1 (magenta, **g**) docking to the *M. maripaludis* McrA (green), B (cyan), G (yellow) complex. The protein subunits are presented in cartoon, and F_430 and CoB-SH are depicted in stick models. Only one active site composed of the *M. maripaludis* McrA (green), A′ (yellow), B (cyan), and the ANME McrG (magenta) subunits and one F_430 are shown for clarity. The amino acids within 8 Å surrounding F_430 are shown as surface representation. **h** The ten amino acids of McrG_ANME-2_HR1 within 8 Å surrounding F_430 are labeled on the left and shown in stick models.

8 Å surrounding F_430 (Fig. 6h), consistent with the reported F_430 binding sites (Supplementary Fig. S4)[26,51]. By contrast, the docking of McrG_ANME-1_BS with *M. maripaludis* MCR had poor quality (I_rmsd = 5.869 Å) (Fig. 6e), and an interaction surface within 8 Å was not observed between McrG_ANME-1_BS and F_430 (Fig. 6f). These simulations agreed with the experimental data that ANME-2 and methanococcal MCR subunits can form a hybrid complex whereas ANME-1 MCR only purified as a homogenous complex.

## Discussion

In this study, a robust MCR expression system in *M. maripaludis* with a Flag-Strep_2 tag on the N-terminal of McrG under the control of the P*pst* promoter was developed. This system increased MCR expression two- to threefold over other promoters, such as the constitutive P*hmv*A, and allowed rapid purification of tagged holo-MCRs. Using this system, recombinant

MCRs were fully assembled with coenzyme F_430 and contained the PTMs present in the *M. maripaludis* MCR. Although the yield of ANME MCR is currently lower than methanogenic MCR, this study set an important step for biochemical and mechanistic studies of MCR homologs from uncultured archaea.

Our heterologous expression provided mechanistic insights into MCR assembly. Previously, we proposed an ordered assembly model for the production of MCR in *M. maripaludis*[36]. In this model, transcription of the *mcr* operon was concurrent with translation and assembly of the subunits into the mature holoenzyme with correct PTMs. Although the experiments reported here were not designed to provide critical tests of this model, they do suggest that the original model was too simplistic. For instance, the genes for the accessory proteins McrC and McrD were not required in the operon for full MCR assembly with F_430 insertion and correct PTMs as the previous model suggested. However, complex production level was reduced by ~60% in *M. maripaludis* when *mcrC* and/or *mcrD* were absent

from the operon. Moreover, pull-down experiments demonstrated that these accessory proteins can function in *trans*. These results suggest that operon structure may play an important role but is not essential for MCR assembly, consistent with the lack of *mcrC/D* in most ANME and ANKA *mcr* operons. Secondly, although chimerism for the McrA and McrB subunits was small as the ordered assembly model predicted, McrG was highly chimeric and assembled into complexes from distinct origins. It is possible that during assembly addition of McrG is less selective than McrA and McrB. For instance, McrG could be the last subunit joining the McrAB complex, bringing in coenzyme $F_{430}$. Alternatively, McrG may be mobile. After initially assembling in MCR as predicted by the ordered assembly model, it is exchanged between mature native and recombinant MCRs. Further experimentation will be necessary to distinguish between these and other hypotheses.

Our protein characterization and structural modeling demonstrated that ANME-2 and methanococcal MCR subunits can form a hybrid complex, suggesting that they are structurally and biochemically more similar to each other than originally thought. ANME-2 archaea belong to the *Methanosarcinales* order, which is phylogenetically distinct from the *Methanococcales* order. Although *M. marburgensis* MCR has been shown to catalyze reversible $CH_4$ production/$CH_4$ oxidation reactions in *vitro*[20] and ANME archaea were found to be dominant in some methanogenic sediments[52–55], such reversibility has not yet been proven under physiological conditions. Our results provided further evidence that methanogenesis and methanotrophy are regulated by thermodynamic drivers rather than intrinsic differences of MCRs in methanogens and ANME.

## Methods

**Bioinformatics analyses of *mcr* genes**. Most archaeal genomes were retrieved from the Genome Taxonomy Database (GTDB, https://gtdb.ecogenomic.org/). The ANME-1_BS genome (accession no. FP565147) was from the NCBI Nucleotide database. The *Ca*. Ethanoperedens thermophilum E50 genome was downloaded from the GenBank assembly (accession no. GCA_905171685.1)[24]. The taxonomic assignments were made consistent with GTDB release 95 using analysis of relative evolutionary divergence[37]. The genomes were analyzed with CheckM[56] for their completeness and contamination. A set of 1070 genomes with completeness >80% and contamination <10% were selected for this study. The *mcr* genes were searched using tblastn with *Methanococcus maripaludis* strain S2 and *Methanosarcina acetivorans* strain C2A MCRs as the query sequences. The identified subject sequences were excluded if their Expect (E) values were larger than 1e-5. Operon structures were recognized if the identified *mcr* genes were located on the same contig and strand and the distance between two adjacent genes was smaller than 250 bp.

The genome taxonomy tree was constructed using Graphical Phylogenetic Analysis[57], a Python-based command-line tree-drawing tool developed by the Huttentower laboratory. Genome taxonomy and naming are consistent with the GTDB release 95[37]. The patterns of the discovered operons were represented by multiple rings outside the circular phylogenetic tree. Multiple hits of a single genome indicate multiple copies of *mcr* present in that genome.

**Strains and culture conditions**. The recombinant *M. maripaludis* strains were grown anaerobically at 25 or 37 °C. Cells were cultured in 28-mL aluminum-capped, rubber stopper-sealed tubes with 5 mL of minimal formate medium (McF) or rich formate medium (McFc, McF plus 2 g L$^{-1}$ of Casamino acids)[58]. The headspace was 104 kPa of $N_2/CO_2$ (4:1, vol/vol). The 1.5-L cultures were grown in a formate-based medium McF with limiting (80 µM) potassium phosphate dibasic ($K_2HPO_4$). The inoculum was pre-grown in 5 mL of McFc and then transferred into McF with 80 µM $K_2HPO_4$ before inoculating 4% volume into the experimental cultures. Puromycin (1.25 or 2.5 µg mL$^{-1}$) was added when necessary. Prior to inoculation, 3 mM sodium sulfide was added as the sulfur source.

**Plasmids and recombinant strain construction**. The *mcr* genes were cloned into the pMEV4 shuttle vector with a Flag-Strep$_2$ tag under the control of 93-bp P*pst* promoter[47]. For protein expression, the plasmids were transformed into *M. maripaludis* S0001[59] using the polyethylene glycol mediated transformation method[60]. The plasmids were maintained in the recombinant *M. maripaludis* strains by adding 1.25 or 2.5 µg mL$^{-1}$ puromycin to the medium. The colonies of the selected transformants were verified by PCR and sequencing.

**Expression and purification of recombinant MCRs**. *M. maripaludis* cultures expressing recombinant MCR$_{aeo}$ or MCR$_{mar}$ were grown at 37 °C in 1.5 L McF with 80 µM $K_2HPO_4$ until they reached an absorbance at 600 nm of 0.5–0.7. Protein purification was performed under aerobic conditions. The cells were harvested by centrifugation at 17,700 × *g* for 20 min at 4 °C and then resuspended in 5 mL binding buffer containing 100 mM Tris-HCl (pH 7.6), 150 mM NaCl, and Protease Inhibitor Cocktail Tablets (Roche, New York, MO, USA). Cells were lysed by sonication (Fisher Scientific Sonic Dismembrator Model 100) using a cycle of 5 s ON/OFF with the output set at 4 and the duty cycle set at 40% for 20 min on ice. The cell lysate was centrifuged at 17,700 × *g* for 20 min at 4 °C to remove cell debris. The supernatant fraction was loaded on a column containing 1 mL of Strep-Tactin Superflow Plus resin (IBA Lifesciences, Göttingen, Germany) equilibrated with the binding buffer. The column was washed with the binding buffer, and the proteins were eluted with the elution buffer containing 100 mM Tris-HCl (pH 7.6), 150 mM NaCl, and 2.5 mM desthiobiotin. The eluted fractions were desalted and concentrated with an Amicon Ultra centrifugal filter (Millipore, 10-kDa molecular weight cutoff) by centrifugation at 5000 × *g* for 20 min at 4 °C and supplemented with 4 mL buffer A containing 50 mM Tris-HCl (pH 7.6). The protein solution was then loaded on a Q-Sepharose XK16 anion-exchange column equilibrated with buffer A using an NGC liquid chromatography system (Bio-Rad). The protein was eluted with a linear gradient of 0% to 100% buffer B (Buffer A plus 1 M NaCl). The colored fractions containing coenzyme $F_{430}$ were pooled and concentrated to 1 mL using a 10-kDa cutoff centrifugal filter. Protein concentrations were determined with a Pierce BCA protein assay kit (Thermo Fisher Scientific).

*M. maripaludis* cultures expressing ANME MCR were grown at 25 °C in 1.5 L McF with 80 µM $K_2HPO_4$ until they reached an absorbance at 600 nm of 0.5–0.7. The purification of ANME MCR was performed using the Strep-Tactin affinity chromatography as described above. After concentrating with the 10-kDa cutoff centrifugal filter, Strep-Tactin XT magnetic beads (IBA Lifesciences, Göttingen, Germany) were used to further purify ANME MCRs according to the manufacturer's instructions.

**$F_{430}$ extraction and quantification**. For quantification of protein containing $F_{430}$, UV–visible absorption spectra were recorded on an Agilent Cary 60 UV–Vis spectrometer (Agilent Technologies Inc., Palo Alto, CA, USA) with samples in a 10 mm-pathlength quartz cuvette. The amount of $F_{430}$ was calculated with a molar extinction coefficient $\varepsilon = 22,500$ M$^{-1}$ cm$^{-1}$ at 430 nm.

For $F_{430}$ extraction, the purified MCR was treated with an equal volume of 100% methanol, and the precipitated proteins were removed by centrifugation at 17,000 × *g* for 5 min. The supernatant containing free $F_{430}$ was subjected to high-performance liquid chromatography (HPLC) analysis using a C18 column (4.6 × 100 mm, 3.5 µm) on an Agilent 1260 Infinity System equipped with a Diode Array Detector (DAD) VL+ as described previously[41] with minor modifications. Solvent A was 10% acetonitrile and 0.5% formic acid in water and solvent B was 0.5% formic acid in 100% acetonitrile. The flow rate was 0.5 mL min$^{-1}$, and the injection volume was 30 µL. The linear gradient elution was employed in the following manner: 0–10% B over 25 min, 10–100% B over 5 min. The spectrum was recorded from 260 to 640 nm. Quantification was based upon the standard curve constructed with authentic $F_{430}$ (Supplementary Fig. S1).

**Protein mass spectrometry**. The purified MCR subunits were separated on precast 4–20% SDS-PAGE gels (Bio-Rad) and then stained with AquaStain (Bulldog Bio) or with the Pierce™ Silver Stain Kit (Thermo Fisher Scientific). For In-gel trypsin digestion, the gel bands were sliced into small pieces and then rinsed twice with 50% acetonitrile/20 mM ammonium bicarbonate (~pH 7.5–8). The gel pieces were dehydrated by adding 100% of acetonitrile and dried in a SpeedVac. Various amounts of a trypsin solution (0.01 µg µL$^{-1}$ in 20 mM ammonium bicarbonate) were added until the gel pieces totally absorbed the solution. The samples were incubated at 37 °C overnight. The tryptic peptides were extracted from gel pieces by incubating twice with 50% acetonitrile/0.1% formic acid. The extracts were dried by a SpeedVac. A similar protocol was used for in-gel pepsin digestion. After the gel pieces were rinsed with 50% acetonitrile/20 mM ammonium bicarbonate to destain, the gel pieces were rinsed with 0.1% formic acid twice before dehydration with 100% acetonitrile. Sufficient pepsin solution (Promega, 0.02 mg mL$^{-1}$ in 0.04 M HCl) was added to cover the gel pieces. The samples were digested at 37 °C overnight (16–18 h). The peptides were extracted with 50% acetonitrile in water. For in-solution trypsin digestion, samples were diluted with 20 mM ammonium bicarbonate to 0.5–1 g L$^{-1}$ and supplemented with dithiothreitol at a final concentration of 10 mM. The samples were incubated at 100 °C for 5–10 min and allowed to cool to room temperature. The proteins were then digested with trypsin at the ratio of 50:1, protein to trypsin (w/w) overnight at 37 °C. The sample was then dried in a vacufuge.

For protein identification, the peptide mass fingerprinting (PMF) of gel bands were analyzed by a Bruker Autoflex Matrix-Assisted Laser Desorption Ionization (MALDI) Time-of-Flight (TOF) mass spectrometer. The matrix compound 2,5 dihydroxybenzoic acid (2,5-DHBA) was dissolved in 50% methanol to make a ~10 g L$^{-1}$ solution. About 0.5–1 µL of the matrix solution and sample solutions ($F_{430}$ and Tryptic peptides) were mixed and deposited on a metal plate and allowed to dry completely.

For PTM analyses and quantifications of the relative abundance of chimeric MCR subunits, the liquid chromatography with tandem mass spectrometry (LC-MS/MS) analyses were performed on a Thermo Fisher LTQ Orbitrap Elite Mass Spectrometer coupled with a Proxeon Easy NanoLC system (Waltham, MA). The peptides were resuspended in 0.1% formic acid and then loaded into a reversed-phase column (self-packed column/emitter with 200 Å 5 μM Bruker MagicAQ C18 resin), then directly eluted into the mass spectrometer. Briefly, the two-buffer gradient elution (0.1% formic acid as buffer A and 99.9% acetonitrile with 0.1% formic acid as buffer B) started with 5% B, held at 5% B for 2 min, then increased to 25% B in 60 min, to 40% B in 10 min, and to 95% B in 10 min. The data-dependent acquisition (DDA) method was used to acquire MS data. A survey MS scan was acquired first, and then the top 5 ions in the MS scan were selected for the following CID and HCD MS/MS analysis. Both MS and MS/MS scans were acquired by Orbitrap at the resolutions of 120,000 and 30,000, respectively. Data were acquired using Xcalibur software (version 2.2, Thermo Fisher Scientific). The protein identification and modification characterization were performed using Thermo Proteome Discoverer (version 1.3/1.4/2.2) with Mascot (Matrix Science) or SEQUEST (Thermo) programs. The spectra of modified peptides were inspected further to verify the accuracy of the assignments. For quantification of the relative abundance, the chromatographic peak areas of the identified peptides belonging to the same MCR subunit were extracted and combined. The relative abundance was calculated by direct comparison of the combined peak areas (Supplemental Data 2).

For intact protein mass spectrometry, the purified MCR was prepared at 10 μM concentration in 200 mM ammonium acetate (pH 7.6). Spectra were acquired using a 12 T Bruker Solarix FT-ICR-MS instrument. The sample was introduced into the instrument via nano-electrospray with 30-μm fused silica emitter tips (New Objective, Inc.) at a flow rate of 300 nL min$^{-1}$. Ion optics were optimized for the transmission of high-m/z ions by setting all RF frequencies to their lowest values (octupole 2.0 MHz, collision cell 1.4 MHz, and transfer 1.0 MHz) and using a time of flight of 3 ms. Spectra were smoothed using a Savitzky–Golay filter in Bruker DataAnalysis. Charge state assignments and deconvoluted masses were determined manually by standard techniques. Briefly, for each of the measured $m/z$ ratios, the mass was calculated as mass $= (m/z)z$-$z$, where $z$ is the charge. For Complex I, the peaks were attributed to $z$ of 28–32. For Complex II, the peaks were attributed to $z$ of 28–31. The reported masses are then the averages of the values for each $m/z$ ratio, and the standard deviation was calculated from the variation of the calculated masses.

**Western blotting**. After the separation of proteins on precast 4–20% SDS-PAGE gels (Bio-Rad), they were transferred onto methanol-activated polyvinylidene difluoride (PVDF) membranes. Nonspecific binding was blocked with 5% milk in phosphate-buffered saline and 0.1% Tween 20 (PBST) for 1.5 h at room temperature. The PVDF membranes were then incubated with primary antibodies (1:1000 dilution; catalog no. A8592, Sigma-Aldrich) against the FLAG tag for 1.5 h at room temperature and washed three times for 15 min with PBST. Then PVDF membranes were developed using the Western horseradish peroxidase (HRP) substrate for enhanced chemiluminescent detection (catalog no. 32132; Thermo Fisher Scientific). As reported before[47], the relative intensity of each immunoreactive band was estimated with ImageJ, where a linear response was confirmed over the range used.

**Quantitative real-time PCR (qRT-PCR)**. The RNA extraction and qRT-PCR were performed as described[47]. The primers were designed based on the DNA sequences of *M. maripaludis mcrA* and ANME-1_G37 *mcrB* genes using Thermo Fisher Primer Express software v3.0.1, with a melting temperature of 60 °C. The primer sequences were as follows: 5′-GTTCACCCTTCCCTTGCATG-3′ (*M. maripaludis mcrA*-forward); 5′-TGTTGATGTCGATTAAGAATCTGCT-3′ (*M. maripaludis mcrA*-reverse); 5′-TCGTTAACCTGACCATTCGGA-3′ (G37 *mcrB*-forward); 5′-CCGCGGATTACCATTCCTTT-3′ (G37 *mcrB*-reverse). Standard curves were created with 10-fold serial dilutions between $10^9$ and $10^5$ copies per reaction. For qRT-PCR, 30 ng of total RNA was used for each PCR reaction. All samples fit within the standard curve, and the amplification efficiency was 97% with $R^2$ of 0.99.

**Protein structural modeling**. RosettaCM, an improved method for comparative modeling, was used for obtaining the optimized structures[48]. The *M. maripaludis* MCR was modeled using MCRs from *Methanopyrus kandleri* (PDB code 1E6V), *Methanosarcina barkeri* (PDB code 1E6Y), *Methanothermobacter thermautotrophicus* (PDB code 1HBM), *Methanothermobacter marburgensis* (PDB code 3POT, 5A8R), *Methanothermobacter wolfeii* (PDB code 5A8K, 5A8W), an uncultured archaeon (PDB code 3SQG), *Methanothermococcus thermolithotrophicus* (PDB code 5N1Q), *Methanotorris formicicus* (PDB code 5N28), and *Methermicoccus shengliensis* (PDB code 7NKG) as templates. The models of the ANME-2_HR1 McrG was based on MCRs from *Methanosarcina barkeri* MCR (PDB code 1E6Y), *Methanothermobacter thermautotrophicus* (PDB code 1HBM), *Methanothermobacter marburgensis* (PDB code 3POT, 5A8R), *Methanothermobacter wolfeii* (PDB code 5A8K, 5A8W), and *Methermicoccus shengliensis* (PDB code 7NKG) as templates. Local docking searches were carried out using RosettaDock (Rosetta version 3.12)[49,50], and a starting structure was generated from the pre-packed input structure with random Gaussian perturbations of 3 Å for translation and 8° for rotation ("-docking:dock_pert 3 8"). The default rotamer library was appended with extra chi1 and chi2 aromatic rotamers ("-ex1 -ex2aro"). The defined docking partners by chain IDs

made sure the ANME McrG was moved around the trimer of *M. maripaludis* McrA, B, and A'. ("-docking:partners ABD_C"). About 60,000 models were produced in each docking run ("-nstruct 60000").

A number of measurements of structural accuracy are regularly used to measure docking performance, as defined by the Critical Assessment of Protein Interactions (CAPRI) evaluators[61]. I_rmsd is defined as the root mean squared deviation (RMSD) of the heavy atoms in the interface residues after superposition of those same residues, where the interface is defined as all residues with an intermolecular distance of at most 8 Å. We classified our docking results based on whether they achieved a docking funnel. According to the CAPRI-defined criteria, a model with I_rmsd < 1.0 Å was considered high quality, 1.0 Å < I_rmsd < 2.0 Å was considered medium quality, and 2.0 Å < I_rmsd < 4.0 Å was considered acceptable quality. The structures were viewed and adjusted in PyMOL (The PyMOL Molecular Graphics System, version 2.5, Schrödinger).

**Statistics and reproducibility**. The number of samples for each experiment is provided in the figure legends and the Supplementary Information. Statistical analyses were performed using GraphPad Prism 9 software, and data are presented as the mean ± standard deviation (SD).

**Reporting summary**. Further information on research design is available in the Nature Research Reporting Summary linked to this article.

## Data availability

All data generated or analyzed during this study are available within the paper and Supplementary Information files. Full-length uncropped original western blots and gels used in the manuscript are shown in Supplementary Fig. S6. Raw mass spectrometry data for quantification of the relative abundance are included in Supplementary Data 2. Raw data for figure plotting are included in Supplementary Data 3. The plasmids used in this study can be accessed in Addgene under accession codes 192763, 192764, 192765, 192766, 192767, 192768, 192769, 192770, 192771, 192772, 192773, 192774, 192775, 192776, 192777, 192778.

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

## Acknowledgements

We thank Steven Mansoorabadi (Auburn University) for valuable discussions and Saw Kyin (Princeton Proteomics & Mass Spectrometry Facility) for some of the protein mass spectrometry analyses. The plasmids construction and bioinformatics analysis were supported by a grant from the ExxonMobil Research and Engineering Company to W.B.W. and E.C.D., and the characterization of purified proteins was supported by the grant from the U.S. Department of Energy (DE-SC0018028) to W.B.W. and E.C.D. The native mass spectrometry analysis was supported by an instrumentation grant 1S10 OD025118. Thermo Orbitrap Elite and FT-ICR were further supported by grants S10RR028859 and S10OD025118 to the Proteomics and Mass Spectrometry Facility at the University of Georgia.

## Author contributions

N.S., Y.L., E.C.D., and W.B.W. designed the research. N.S. performed genetics and protein purification and characterization experiments. C.W.C. performed most of the protein mass spectrometry analyses. Y.F. and S.M.B. did bioinformatics analyses and protein structural modeling. S.Y. performed the McrC$_{mar}$ pull-down experiment. R.V.W. performed the intact protein mass spectrometry in the laboratory of I.J.A. I.J.D. performed the qRT-PCR experiments. All authors analyzed the data. N.S., Y.L., and W.B.W. wrote the manuscript, which was edited by all authors.

## Competing interests

The authors declare no competing interests.
