## [Peer Review File · Communications Biology]

Reviewers' comments:

Reviewer #1 (Remarks to the Author):

Shao et al developed in *Methanococcus mariplaudis* a platform to recombinantly express *mcr* genes, encoding methyl-CoM reductase, from methanogenic and methanotrophic archaea. Specifically, using Flag-Strep-tagged McrC, the *mcrBDCGA* gene cluster of *M. aerolicus* was heterologously expressed using a new, low-Pi-induced promoter (P_{pst}) as proof of concept and found the assemble into a MCR protein complex of expected size and cofactor (F430) content. Substantial and careful work went into developing this very useful platform and to identify cross-incorporation of host Mcr subunits. This platform was used to express heterologously *mcr* genes from two ANME-1, four ANME-2 and one ECR MCR operons, that were identified in metagenomes. Differences in MCR complex assembly were found as expected based on subunit composition, operon structure, and predicted structural models.

Understanding the mechanistic basis of anaerobic methane and alkane activation and oxidation via the MCR complex and pathway of reverse methanogenesis is one of the last missing big puzzles in microbial biology. This experimental platform has the promise to answer key mechanistic questions on anaerobic methane/alkane activation in the absence of new isolates, which are very difficult to obtain. The experiments were carefully conducted and thoughtfully characterized. I appreciate the substantial work for developing this platform. The carefully-written manuscript would carry significantly more weight if the expressed MCRs from at least one methanogenic or methanotrophic genomes would have been characterized functionally.

Minor comments:

1. Please briefly summarize the functions of the different MCR subunits, including of McrC and D.
2. What is the surmised catalytic consequence of a lack of McrC and McrD?

Unfortunately, there were no line numberings so the authors need to look up the corresponding sentences in the manuscript.

3. remain 'unculturable' should read 'uncultured'
4. 'opposite direction of biosynthesis' This might be an issue under debate. I suggest to use 'biosynthesis' for anabolic, not catabolic processes.
5. 'physiological adaptations', should read 'physiological acclimations
6. Title 'Operon structure affects heterologous expression levels of MCR'. Since expression studies were only performed in *M. mariplaudis*, this constraint should be added in the title.
7. 28 -ml 'aluminum-sealed tubes', I assume should read 'aluminum-capped, rubber stopper-sealed tubes'

Reviewer #2 (Remarks to the Author):

This is an extremely high quality manuscript, both with regard to writing quality, dataset quality and figures, that represents a significant advance in the field, by development of a new expression system for a globally and evolutionarily important anaerobic protein complex that has been notoriously difficult to study. In addition to the importance of the method, the paper also provides important insights about the importance of accessory proteins for complex assembly and structural and

biochemical similarities between different MCRs. The supplemental dataset contains a meticulously documented list of all protein accession numbers, which is enormously helpful for other researchers. I have only a few tiny suggestions for minor grammatical improvements.

Specific comments:

First paragraph of results: "interspersed with lineages that do not share these genes" - please clarify this phrasing, as all the lineages on the tree must share these genes in order to be included on the tree

Throughout: sometimes the subscript "ae o" has a space and other times it does not. Ensure the same nomenclature is used consistently throughout.

Reviewer #3 (Remarks to the Author):

In this manuscript, Shao et al. perform heterologous expression of MCR derived from a methanogen (*Methanococcus aeolicus*), ANME and ANKA in the model methanogen, *Methanococcus maripaludis*. The manuscript shows that strains expressing foreign MCR variants produce chimeric MCR protein comprised of subunits of the foreign MCR and the native MCR. The paper also shows that assembly of the chimeric MCR can occur when *mcrC* and *mcrD* are expressed in trans.

Overall, the manuscript contains some important preliminary evidence that can ultimately lead to the production of a functional MCR from ANME and ANKA but stops short of generating any novel insights about MCR. Many experiments lack important controls and the results are overinterpreted often without any attention to alternate hypotheses that are equally consistent with the outcome of the experiment. As such, substantial work is required to bolster some of the claims made in the manuscript as outlined below.

Introduction

The introduction is sparse and does not provide any background on many recent studies on heterologous expression of MCR from other methanogens (such as Lyu et al. *J. Bacteriology* 2018) or ANME (Soo et al. *Microbial Cell Factories* 2016). In fact, some of these studies were conducted by the same group of researchers (see Lyu et al. *J. Bacteriology* 2018) with conclusions that run counter to the ones drawn in this work. It is imperative that the introduction be rewritten to provide more background and motivation for the current work.

It should also be noted that PTMs have been shown to fine-tune the stability of MCR but not its activity.

Results

The first section of the results titled "MCRs are widespread and diverse in archaea" does not provide any information that has not been published before. This section is more appropriate for the introduction.

'the widespread distribution of *mcr* genes in archaea suggests that methane metabolism is an ancient trait likely present at the root of the archaeal tree'. The phylogenetic analyses conducted in this manuscript do not provide any support to this hypothesis and the statement should be removed.

'the lack of *mcrD* in ANME-1 and ANKA may correlate with modified nickel-containing cofactor F430'. This statement is highly speculative and should be removed.

'The Phmv promoter and Ppst promoter yielded.....2.4 and 5.8% of total protein, respectively'. The authors should provide data for this important result and also briefly discuss how this quantification was conducted.

'an HPLC-based analyses'. Either remove this phrase or provide more detail on how this HPLC based analyses were used to quantify the amount of F430 in the MCR complex.

Re: PTM analyses. it is unclear how the authors concluded that their heterologous system resulted in correct PTMs of the *M. aeolicus* MCR as (to the best of my knowledge) the PTMs in MCR derived from *M. aeolicus* have never been characterized.

'Intact protein mass spectrometry determined that complex I and II had molecular mass of 288.4 and 283.8 kDa respectively (Figure 3c).' It isn't clear how these molecular masses were calculated based on the data shown in figure 3c. Please clarify.

'These results indicate that McrG was the most mobile subunit of MCR and readily binds McrA and B subunits from a different origin'. While this hypothesis is plausible, previous work by these researchers (Lyu et al. 2018) contradicts this model to show that only subunits encoded in an operon form a complex. Since their previous study is not consistent with this one, the authors should either provide some clarification for the underlying differences or refrain from making any strong conclusions about the assembly of MCR altogether.

Re: 3 fold lower MCR production in the *mcrC* and *mcrD* KO mutants. While it is plausible that the McrC/D proteins from the native host in trans are less efficient than those in cis, there are other hypotheses that are equally consistent with these observations. For instance, removing *mcrC* or *mcrD* could change the stability of the *mcr* transcript leading to lower protein production. Or, these mutants have altered RBS's for genes downstream of *mcrD* or *mcrC*. Or, the native host McrC/D proteins do not work with 100% efficiency with the McrABG from *M. aeolicus*, so this effect is due to host machinery not being as effective as the heterologous machinery, irrespective of the cis/trans issue. The authors need to control for some of these possibilities before concluding that the McrC and McrD from the native host acting in trans are less efficient. Alternately, these results and their discussion can be eliminated from the manuscript altogether.

Re: the effect of temperature on heterologous expression of ANME/ANKA MCR. It is common practice to lower the temperature to increase production of heterologous proteins in *E. coli*. Similarly, increased production of ANME/ANKA MCR at lower temperatures in *M. maripaludis* might have little to do with the temperature stability of the complex and have more to do with host-specific factors like proteolytic enzymes etc. that are less active at low temperatures. It is also worth noting that 25 C is *VERY* hot for ANME as these organisms thrive in environments that are much colder (<10 C). The authors must perform appropriate controls to justify the temperature specific effects on the assembly of ANME/ANKA MCR or remove these data altogether.

Discussion

'Although the yield of ANME MCR is currently lower than methanogenic MCR, this study set an important step for biochemical and mechanistic studies of MCR homologs from uncultured archaea'. There are no results in this manuscript that justify this statement. One could argue that the production of chimeric MCRs as shown in this manuscript prevents the study of MCR homologs from other organisms. Additionally, this manuscript does not provide any evidence that the MCR expressed from ANME or ANKA are active in *M. maripaludis*. This statement should be removed or corroborated with additional experiments described above.

As mentioned above, the manuscript provides very little evidence that McrG is that is recruited after McrAB assemble or is involved in bringing in F430. The discussion should refrain from building a model based on scant evidence.

Reviewer: 1

Shao et al developed in *Methanococcus marislaudis* a platform to recombinantly express mcr genes, encoding methyl-CoM reductase, from methanogenic and methanotrophic archaea. Specifically, using Flag-Strep-tagged McrC, the mcrBDCGA gene cluster of *M. aerolicus* was heterologously expressed using a new, low-Pi-induced promoter (Ppst) as proof of concept and found the assemble into a MCR protein complex of expected size and cofactor (F430) content. Substantial and careful work went into developing this very useful platform and to identify cross-incorporation of host Mcr subunits. This platform was used to express heterologously mcr genes from two ANME-1, four ANME-2 and one ECR MCR operons, that were identified in metagenomes. Differences in MCR complex assembly were found as expected based on subunit composition, operon structure, and predicted structural models.

Understanding the mechanistic basis of anaerobic methane and alkane activation and oxidation via the MCR complex and pathway of reverse methanogenesis is one of the last missing big puzzles in microbial biology. This experimental platform has the promise to answer key mechanistic questions on anaerobic methane/alkane activation in the absence of new isolates, which are very difficult to obtain. The experiments were carefully conducted and thoughtfully characterized. I appreciate the substantial work for developing this platform. The carefully-written manuscript would carry significantly more weight if the expressed MCRs from at least one methanogenic or methanotrophic genomes would have been characterized functionally.

We appreciate the reviewer's positive comments. We agree that the functional characterization of these MCRs is important, and it is part of our continuing investigations and outside the scope of this study.

Minor comments:

1. Please briefly summarize the functions of the different MCR subunits, including of McrC and D.
2. What is the surmised catalytic consequence of a lack of McrC and McrD?

Unfortunately, there were no line numberings so the authors need to look up the corresponding sentences in the manuscript.

We apologize for this. Line numberings are present in our copies so we assume that they were lost during formation of the pdfs.

More explanation for the role of individual subunits has been added to make the story more understandable. The roles of Mcr A, B and G are described further on lines 64-67. The roles of McrC and D are discussed on lines 95-98.

Lines 64-67:

"The MCR complex is composed of a dimer of heterotrimers $(\alpha\beta\gamma)_2$ with a molecule of the Ni-containing tetrapyrrole coenzyme F_{430} in each of the two active sites²⁵. Each F_{430} is deeply

buried within the protein complex and only accessible from the outside by a 50 Å channel formed from multiple subunits, McrA, A', B, and G or McrA', A, B', and G'^{26, 27}."

Lines 95-98:

"In addition to the structural genes, many *mcr* operons encoded two accessory proteins, McrC and McrD. While the roles of McrC and D are not well characterized, McrC has been shown to participate in the MCR activation complex²⁹ and McrD may facilitate addition of coenzyme F₄₃₀ to the complex⁴¹."

3. remain 'unculturable' should read 'uncultured' Change made, Line 43

4. 'opposite direction of biosynthesis' This might be an issue under debate. I suggest to use 'biosynthesis' for anabolic, not catabolic processes.

Line 46

We changed this to "methane formation" to be clearer. Our concern is that anabolism/catabolism is ambiguous in this context because both reactions are anaerobic respirations. Thus, they are functionally equivalent.

5. 'physiological adaptations', should read 'physiological acclimations' Change made, Line 112

6. Title 'Operon structure affects heterologous expression levels of MCR'. Since expression studies were only performed in *M. maripaludis*, this constraint should be added in the title. Change made, Line 161

7. 28 -ml 'aluminum-sealed tubes', I assume should read 'aluminum-capped, rubber stopper-sealed tubes' Change made, Line 301

Reviewer: 2

This is an extremely high quality manuscript, both with regard to writing quality, dataset quality and figures, that represents a significant advance in the field, by development of a new expression system for a globally and evolutionarily important anaerobic protein complex that has been notoriously difficult to study. In addition to the importance of the method, the paper also provides important insights about the importance of accessory proteins for complex assembly and structural and biochemical similarities between different MCRs. The supplemental dataset contains a meticulously documented list of all protein accession numbers, which is enormously helpful for other researchers. I have only a few tiny suggestions for minor grammatical improvements.

We appreciate the reviewer's positive comments.

Specific comments:

First paragraph of results: "interspersed with lineages that do not share these genes" - please clarify this phrasing, as all the lineages on the tree must share these genes in order to be included on the tree

We rewrote this clarify our statements. The tree is based upon all 1,070 genomes and not the Mcr genes. We now phrase this as:

"A total of 307 genomes contained all three of the genes (*mcrA*, *mcrB*, and *mcrG*) necessary to encode the MCR subunits (Supplementary Table 1). In the rank-normalized phylogenetic tree based upon all 1,070 genomes³⁷, these *mcr*-containing archaea included methanogens, ANME-1, ANME-2, ANKA, and other archaea of unknown metabolic types and were interspersed with lineages that do not share these genes (Fig. 1)," [underline is new text]

Throughout: sometimes the subscript "ae o" has a space and other times it does not. Ensure the same nomenclature is used consistently throughout.

Thanks for noticing this. The space inside of the subscript "ae o" was produced when transferred from word to pdf. We will check this in future drafts.

Reviewer: 3

In this manuscript, Shao et al. perform heterologous expression of MCR derived from a methanogen (*Methanococcus aeolicus*), ANME and ANKA in the model methanogen, *Methanococcus maripaludis*. The manuscript shows that strains expressing foreign MCR variants produce chimeric MCR protein comprised of subunits of the foreign MCR and the native MCR. The paper also shows that assembly of the chimeric MCR can occur when *mcrC* and *mcrD* are expressed in trans.

Overall, the manuscript contains some important preliminary evidence that can ultimately lead to the production of a functional MCR from ANME and ANKA but stops short of generating any novel insights about MCR. Many experiments lack important controls and the results are overinterpreted often without any attention to alternate hypotheses that are equally consistent with the outcome of the experiment. As such, substantial work is required to bolster some of the claims made in the manuscript as outlined below.

We have added more explanation to the manuscript to address the reviewer's concerns.

Introduction

The introduction is sparse and does not provide any background on many recent studies on heterologous expression of MCR from other methanogens (such as Lyu et al. *J. Bacteriology* 2018) or ANME (Soo et al. *Microbial Cell Factories* 2016). In fact, some of these studies were conducted by the same group of researchers (see Lyu et al. *J. Bacteriology* 2018) with conclusions that run counter to the ones drawn in this work. It is imperative that the introduction be rewritten to provide more background and motivation for the current work.

The papers by Lyu et al and Soo et al are now introduced on Lines 74-79. Lyu et al is discussed in depth in the discussion.

Line 74-79: “Heterologous expression of the genes encoding an ANME-1 MCR in *Methanosarcina acetivorans* stimulated methane oxidation by the recombinant organism, providing further evidence for the role of these enzymes³⁵. Recently the *Methanothermococcus okinawensis* MCR was heterologously expressed in the model methanogen *Methanococcus maripaludis*³⁶. Here, we further developed the heterologous expression of MCRs in *M. maripaludis* that paves the way for studying enzyme complexes from uncultured archaea.”

It should also be noted that PTMs have been shown to fine-tune the stability of MCR but not its activity.

We mentioned that the PTMs “fine-tune the MCR stability and activity” in the original manuscript [Line 71], and cited Nayak et al. PLoS Biol 2020 (ref 30). In our opinion, it is incorrect to state the PTMs do not affect activity, especially in certain methanogens including *M. maripaludis*. Now we have added two more citations that support our statement [ref 31 and 32]. The Lyu et al. 2020 paper showed that the methane formation rate was reduced by 40-60% in the *M. maripaludis* mutant strain lacking the 5-C-(S)-methylarginine PTM.

Results

The first section of the results titled “MCRs are widespread and diverse in archaea” does not provide any information that has not been published before. This section is more appropriate for the introduction.

This first section of the results includes a new bioinformatic analyses describing the distribution, diversity, and operon structures of MCR homologs among archaea which has not been published before. This section provides a comprehensive understanding of the distribution and evolution of MCR. Therefore, this section belongs in the results and not the introduction.

‘the widespread distribution of *mcr* genes in archaea suggests that methane metabolism is an ancient trait likely present at the root of the archaeal tree’. The phylogenetic analyses conducted in this manuscript do not provide any support to this hypothesis and the statement should be removed.

Our phylogenetic tree revealed the wide distribution of MCR-containing archaea and supports the hypothesis that methanogenesis is an ancient process as proposed by the authors cited in ref 32-34. This statement was rewritten to clarify that this hypothesis is attributed to other authors and our data is only supportive.

“The widespread distribution of *mcr* genes in archaea supports the hypothesis that methane metabolism is an ancient trait likely present in the archaeal root³⁸⁻⁴⁰.”

'the lack of *mcrD* in ANME-1 and ANKA may correlate with modified nickel-containing cofactor F430'. This statement is highly speculative and should be removed.

This section has been rewritten to avoid the speculation [Lines 102-106]:

"The lack of *mcrD* homologs in ANME-1 and ANKA genomes may be emblematic of other major differences with the enzymes from methanogens. For instance, they contain modified nickel-containing F₄₃₀ cofactors, e.g. thiomethylated F₄₃₀ from an ANME-1 MCR³³ and dimethylated F₄₃₀ from *Candidatus* *Ethanoperedens thermophilum* MCR³⁴."

'The *Phmv* promoter and *Ppst* promoter yielded.....2.4 and 5.8% of total protein, respectively'. The authors should provide data for this important result and also briefly discuss how this quantification was conducted.

This result was from a previous work, and this section has been rewritten to clarify the source of the information. In addition, a description as to how the Westerns were quantitated were added to the methods.

Lines 122-124: "Based upon Western blotting from a previous study, the *PhmvA* and *Ppst* promoters yielded *M. aeolicus* MCR (MCR_{aeo}) of 2.4 and 5.8 % of total protein, respectively⁴⁷."

'an HPLC-based analyses'. Either remove this phrase or provide more detail on how this HPLC based analyses were used to quantify the amount of F430 in the MCR complex.

The HPLC analysis was described in detail in the original manuscript in the section of the methods entitled "F₄₃₀ extraction and Quantification" and Supplementary Fig. 1. Since this figure was cited in the original manuscript, no change was made.

Re: PTM analyses. it is unclear how the authors concluded that their heterologous system resulted in correct PTMs of the *M. aeolicus* MCR as (to the best of my knowledge) the PTMs in MCR derived from *M. aeolicus* have never been characterized.

The reviewer is correct in that the PTMs of *M. aeolicus* are not known, although it is known for closely related species. This sentence has been rewritten here and in the conclusions:

Lines 139-141: "This indicated that our heterologous expression system resulted in the same PTMs as found for the *M. maripaludis* and closely related *M. okinawensis* MCRs."

Lines 241-242: "Using this system, recombinant MCRs were fully assembled with coenzyme F₄₃₀ and contained the PTMs present in the *M. maripaludis* MCR."

'Intact protein mass spectrometry determined that complex I and II had molecular mass of 288.4 and 283.8 kDa respectively (Figure 3c).' It isn't clear how these molecular masses were calculated based on the data shown in figure 3c. Please clarify.

More explanation is now provided in the methods for the data in Fig. 3c and in supplementary table 3. The new information would allow a knowledgeable investigator to repeat our calculations.

In the methods, the explanation was given:

Lines 416-421: "Charge state assignments and deconvoluted masses were determined manually by standard techniques. Briefly, for each of the measured m/z ratios, the mass was calculated as $\text{mass}=(m/z)z-z$, where z is the charge. For Complex I, the peaks were attributed to z of 28-32. For Complex II, the peaks were attributed to z of 28-31. The reported masses are then the averages of the values for each m/z ratio, and the standard deviation was calculated from variation of the calculated masses."

'These results indicate that McrG was the most mobile subunit of MCR and readily binds McrA and B subunits from a different origin'. While this hypothesis is plausible, previous work by these researchers (Lyu et al. 2018) contradicts this model to show that only subunits encoded in an operon form a complex. Since their previous study is not consistent with this one, the authors should either provide some clarification for the underlying differences or refrain from making any strong conclusions about the assembly of MCR altogether.

This sentence has been rewritten as: "These results indicated that McrG readily binds McrA and B subunits from a different origin." [i.e. deleting the mobility of McrG]. We now compare these results in detail with those of Lyu et al 2018 in the discussion.

Re: 3 fold lower MCR production in the *mcrC* and *mcrD* KO mutants. While it is plausible that the McrC/D proteins from the native host in trans are less efficient than those in cis, there are other hypotheses that are equally consistent with these observations. For instance, removing *mcrC* or *mcrD* could change the stability of the *mcr* transcript leading to lower protein production. Or, these mutants have altered RBS's for genes downstream of *mcrD* or *mcrC*. Or, the native host McrC/D proteins do not work with 100% efficiency with the McrABG from *M. aeolicus*, so this effect is due to host machinery not being as effective as the heterologous machinery, irrespective of the cis/trans issue. The authors need to control for some of these possibilities before concluding that the McrC and McrD from the native host acting in trans are less efficient. Alternately, these results and their discussion can be eliminated from the manuscript altogether.

We have deleted the hypothesis that 'The *M. maripaludis* host proteins may have partially complemented the absence of the *M. aeolicus* McrC and McrD.' And we added the statement that 'although the cause of the reduced expression levels is currently unclear.'

Re: the effect of temperature on heterologous expression of ANME/ANKA MCR. It is common practice to lower the temperature to increase production of heterologous proteins in *E. coli*. Similarly, increased production of ANME/ANKA MCR at lower temperatures in *M. maripaludis* might have little to do with the temperature stability of the complex and have more to do with host-specific factors like proteolytic enzymes etc. that are less active at low temperatures. It is also worth noting that 25 C is *VERY* hot for ANME as these organisms thrive in environments

that are much colder (<10 C). The authors must perform appropriate controls to justify the temperature specific effects on the assembly of ANME/ANKA MCR or remove these data altogether.

We agree with the reviewer that it is common that lower temperature can benefit heterologous protein expressions for multiple reasons. We have added qPCR data to rule out the possibilities of low transcription level/stability of mRNA at 37 °C. The original manuscript had stated the temperature difference between the habitats of the deep sea ANME and *Methanococcus*, so no change was made. However, we added the possibility of 'susceptible to degradation' in Line 211 as the reviewer suggested. We don't agree that additional experiments are needed because the detailed cause of the temperature effect is out of the scope of this study.

Discussion

'Although the yield of ANME MCR is currently lower than methanogenic MCR, this study set an important step for biochemical and mechanistic studies of MCR homologs from uncultured archaea'. There are no results in this manuscript that justify this statement. One could argue that the production of chimeric MCRs as shown in this manuscript prevents the study of MCR homologs from other organisms. Additionally, this manuscript does not provide any evidence that the MCR expressed from ANME or ANKA are active in *M. maripaludis*. This statement should be removed or corroborated with additional experiments described above.

Although the heterologous expression of ANME2 MCR produced chimeric complexes, the expression of ANME1 MCR and ECR indeed produced homogenous complexes. Therefore, this study enabled future studies of MCR from uncultured archaea as we stated. Although their activity was not examined, to our knowledge this is the first successful expression and purification of recombinant MCR of the uncultured archaea.

As mentioned above, the manuscript provides very little evidence that McrG is that is recruited after McrAB assemble or is involved in bringing in F430. The discussion should refrain from building a model based on scant evidence.

This paragraph was extensively rewritten to discuss the hypothesis of Lyu et al and other models for MCR assembly. [Lines 247-265]

REVIEWERS' COMMENTS:

Reviewer #1 (Remarks to the Author):

The modifications introduced in the revised manuscript greatly improved the manuscript.

Reviewer #2 (Remarks to the Author):

All of my comments have been addressed/corrected and I support publication.

Reviewer #3 (Remarks to the Author):

I have looked at the revised manuscript and my concerns have been addressed.